



# 1 Primary emissions versus secondary formation of fine particulate matter in the top polluted
# 2 city, Shijiazhuang, in North China

Ru-Jin Huang[1], Yichen Wang[1], Junji Cao[1], Chunshui Lin[1,2], Jing Duan[1], Qi Chen[3], Yongjie Li[4],
Yifang Gu[1], Jin Yan[1], Wei Xu[1,2], Roman Fröhlich[5], Francesco Canonaco[5], Carlo Bozzetti[5], Jurgita
Ovadnevaite[2], Darius Ceburnis[2], Manjula R. Canagaratna[6], John Jayne[6], Douglas R. Worsnop[6], Imad
El-Haddad[5], André S. H. Prévôt[5], Colin D. O'Dowd[2]
[1]Key Laboratory of Aerosol Chemistry & Physics, State Key Laboratory of Loess and Quaternary
Geology, Institute of Earth Environment, Chinese Academy of Sciences, Xi'an 710061, China
[2]School of Physics and Centre for Climate and Air Pollution Studies, National University of Ireland
Galway, Galway, Ireland
[3]State Key Joint Laboratory of Environmental Simulation and Pollution Control, College of
Environmental Sciences and Engineering, Peking University, Beijing, China
[4]Department of Civil and Environmental Engineering, Faculty of Science and Technology, University
of Macau, Taipa, Macau, China
[5]Laboratory of Atmospheric Chemistry, Paul Scherrer Institute (PSI), 5232 Villigen, Switzerland
[6]Aerodyne Research, Inc., Billerica, MA, USA
*Correspondence to*: R.-J. Huang (rujin.huang@ieecas.cn)
**Abstract.** Particulate matter (PM) pollution is a severe environmental problem in the Beijing-Tianjin-
Hebei (BTH) region in North China. PM studies have been conducted extensively in Beijing, but the
chemical composition, sources, and atmospheric processes of PM are still relatively less known in the
nearby Tianjin and Hebei. In this study, fine PM in urban Shijiazhuang (the capital of Hebei province)
was characterized using an Aerodyne quadrupole aerosol chemical speciation monitor (Q-ACSM)
from 11 January to 18 February in 2014. The average mass concentration of non-refractory submicron
PM (diameter $<1$ μm, NR-PM$_1$) was $178 \pm 101$ μg m$^{-3}$ and composed of 50% organic aerosol (OA),
21% sulfate, 12% nitrate, 11% ammonium, and 6% chloride. Using the Multilinear Engine (ME-2)



receptor model, five OA sources were identified and quantified, including hydrocarbon-like OA from
vehicle emissions (HOA, 13%), cooking OA (COA, 16%), biomass burning OA (BBOA, 17%), coal
combustion OA (CCOA, 27%), and oxygenated OA (OOA, 27%). We found that secondary
formation contributed substantially to PM in episodic events, while primary emissions were dominant
(most significant) on average. The episodic events with the highest NR-PM$_1$ mass range of 300-360
µg m$^{-3}$ showed 55% of secondary species. On the contrary, a campaign-average low OOA fraction
(27%) in OA indicated the importance of primary emissions, and a low sulfur oxidation degree ($F_{SO4}$)
of 0.18 even at RH>90% hinted on insufficient oxidation. These results suggested that in wintertime
Shijiazhuang fine PM was mostly from primary emissions without sufficient atmospheric aging,
indicating opportunities for air quality improvement by mitigating direct emissions. In addition,
secondary inorganic and organic (OOA) species dominated in pollution events with high RH
conditions, most likely due to enhanced aqueous-phase chemistry, while primary organic aerosol
(POA) dominated in pollution events with low RH and stagnant conditions. These results also
highlighted the importance of meteorological conditions for PM pollution in this highly polluted city
in North China.
**1 Introduction**
Particulate pollution in China is a serious environmental problem, influencing air quality, regional and
global climate and human health. Especially during recent winters, large-scale and severe haze
pollution has brought China's particulate pollution at the forefront of world-wide media and evoking
great scientific interest in air pollution studies. Measurements at a number of major cities showed that
the wintertime daily average mass concentration of PM$_{2.5}$ (particulate matter with an aerodynamic
diameter <2.5 µm) are approximately 1-2 orders of magnitude higher than those observed in urban
areas in the US and European countries (Huang et al., 2014). Severe particulate pollution is often
accompanied by extremely poor visibility and poor air quality leading to a sharp increase in
respiratory diseases. Long-term exposure to high levels of particulate pollution is estimated to result
in 1.36 million premature deaths per year in China, ranking the 1$^{st}$ in the world (Lelieveld et al.,

27  2015).



The region of Beijing, Tianjin, and Hebei (BTH) is one of the important city clusters in China, but
also suffers from serious air pollution. Seven cities in this region ranked the top 10 most polluted
cities in China in the year 2014-2015 (http://www.zhb.gov.cn). The urgent need of an air quality
improvement in this region has been recognized by central and local governments as well as the
public, leading to mitigating actions being undertaken by the authorities. In particular, various
emission control measures were implemented in this region to clean Beijing's sky, for example,
during the 2014 Asia-Pacific Economic Cooperation (APEC) summit. These temporal measures
include the odd-even ban on vehicles and shutdowns of factories and construction sites, leading to
serious side effects on daily life and economic growth. Therefore, identification of the major sources
and atmospheric processes producing airborne particles is required for implementing targeted and
optimized emission control strategies.
The first step for quantifying the PM sources requires the measurements of inorganic and organic
tracers and/or mass spectrometric fingerprints of ambient PM samples. This can be realized by the
online ambient measurements using aerosol mass spectrometric (AMS) techniques to determine
aerosol composition (Jimenez et al., 2009; Ng et al., 2011b; Elser et al., 2016b). In particular, the
quadrupole aerosol chemical speciation monitor (Q-ACSM) and recently time-of-flight aerosol
chemical speciation monitor (TOF-ACSM) have been developed for long-term continuous
measurements of the non-refractory submicron aerosols (Ng et al., 2011a; Fröhlich et al., 2013).
Aerosol sources have been successfully identified from the AMS measurements with positive matrix
factorization (PMF) analysis (Ulbrich et al., 2009; Crippa et al., 2013; Elser et al., 2016a). In terms of
Q-ACSM datasets, the use of PMF often fails to resolve sources with similar mass spectral profiles,
e.g. the mixing of cooking organic aerosol with traffic organic aerosol in Nanjing (Zhang et al.,
2015b); or those present in low contributions, e.g. the lack of success in resolving a factor related to
biomass burning in Beijing (Jiang et al., 2015). It was also pointed out that PMF cannot separate the
aerosol sources of temporal covariations driven by low temperature and periods of strong inversions
(Canonaco et al., 2013; Reyes et al., 2016). Several source apportionment studies (in which PMF did
not find optimal results) have utilized the multilinear engine (ME-2) solver, which enables constraint
of the factor profiles/time series, providing a superior separation of the PM sources in Europe (e.g.,



Canonaco et al., 2013; Canonaco et al., 2015; Fröhlich et al., 2015a; Fröhlich et al., 2015b;
Minguillón et al., 2015; Petit et al., 2015; Ripoll et al., 2015; Reyes et al., 2016; Bressi et al., 2016;
Schlag et al., 2016). However, studies using ME-2 to resolve OA sources from the ACSM
measurements are scarce in the BTH region.
Apart from the lack of applications of ME-2 for the OA source apportionment, most of the field
studies have mainly focused on the aerosol pollution in Beijing (Sun et al., 2013; Sun et al., 2014; Sun
et al., 2016; Jiang et al., 2015; Xu et al., 2015; Elser et al., 2016b; Hu et al., 2016a). These and related
studies have clearly shown that Beijing is sensitive to the regional transport of aerosols from its
surrounding areas (Xu et al., 2008; Zhang et al., 2012; Li et al., 2015a). For example, Guo et al.
(2010) estimated that the regional pollutants on average accounted for 69% of $PM_{10}$ and 87% of $PM_{1.8}$
in Beijing during summer, with sulfate, ammonium, and oxalate mostly formed regionally (regional
contributions >87%). Sun et al. (2014) reported that 66% of $NR-PM_1$ was from regional transport in
Beijing during the 2013 winter haze event. Among the surrounding areas of Beijing, the Hebei
province is the main source area leading to high aerosol loadings in Beijing (Chen et al., 2007; Xu et
al., 2008; Lang et al., 2013; Li et al., 2015a).
Shijiazhuang, the capital of Hebei province, is located ~270 km south of Beijing and has a population
approximately half that of Beijing. Zhao et al. (2013a, b) characterized the spatial and seasonal
variations of $PM_{2.5}$ chemical composition in the BTH region, and Shijiazhuang was selected as the
representative of the polluted cities in Hebei province. The off-line analysis results showed that
organic carbon (OC) and elemental carbon (EC) concentrations in Shijiazhuang were lower in the
spring and summer than those in the autumn and winter. The sum of secondary inorganic species
($SO_4^{2-}$, $NO_3^-$, and $NH_4^+$) was highest in the autumn. Yet the temporal profiles of PM composition
cannot be captured by off-line analyses, hindering more detailed study on the sources and formation
of PM. In this work, we present for the first time the 30-minute time resolved $NR-PM_1$ measurements
in Shijiazhuang during the winter heating season. The characteristics of $NR-PM_1$ are analyzed, which
include (1) time series, mass fraction and diurnal variation of $NR-PM_1$ species; (2) multilinear engine
(ME-2)-resolved OA sources and their mass fraction as well as their diurnal variation; and (3) the



characteristics and atmospheric evolution of aerosol composition and sources under different aerosol
loadings and meteorological conditions.
**2 Methods**
**2.1 Sampling site**
Shijiazhuang, the capital of Hebei province, is located ~270 km south of Beijing. In 2014, ~10 million
residents and 2.1 million vehicles were reported in this city. It is often ranked the first in the list of top
10 most polluted cities in China, especially during wintertime heating periods (from 15 November to
15 March next year).  For example, the average concentration of $PM_{2.5}$ was 226.5 µg m$^{-3}$ with the
peak hourly concentration of 933 µg m$^{-3}$ during the 2013/3014 wintertime heating period, largely
exceeding the Chinese air pollution limit of 75 µg m$^{-3}$. In this study, we performed an intensive field
measurement campaign at an urban site in Shijiazhuang to investigate the chemical composition,
sources and atmospheric processes of fine particles. The campaign was carried out from 11 January to
18 February 2014 on the building roof (15 m) of the Institute of Genetics and Developmental Biology,
Chinese Academy of Sciences (38°2'3''N, 114°32'29''E), a site located in a residential-business
mixed zone.
**2.2 Instrumentation**
NR-PM$_1$ was measured with an Aerodyne quadrupole aerosol chemical speciation monitor (Q-
ACSM), which can provide quantitative mass concentration and mass spectra of non-refractory
species including organics, sulfate, nitrate, ammonium, and chloride. The operation principles of Q-
ACSM can be found elsewhere (Ng et al., 2011a). The ambient aerosol was drawn through a Nafion
dryer (Perma Pure PD-50T-24SS) following a URG cyclone (Model: URG-2000-30ED) with a cut-off
size of 2.5 µm to remove coarse particles. The sampling flow was ~3 L min$^{-1}$, of which ~85 mL min$^{-1}$
was isokinetically sampled into the Q-ACSM. The residence time in the sampling tube was ~5 s. The
Q-ACSM was operated with a time resolution of 30 min and scanned from *m/z* 10 to 150 at 200 ms
amu$^{-1}$. Dry  mono-dispersed 300-nm ammonium nitrate and ammonium sulfate particles (selected by a
differential mobility analyzer, DMA, TSI model 3080) were nebulized from a custom-built atomizer
and sampled into the Q-ACSM and a condensation particle counter (CPC, TSI model 3772)





calibrating ionization efficiency (IE). IE can, therefore, be determined by comparing the response factors of Q-ACSM to the mass calculated with the known particle size and the number concentration from CPC.

Ozone ($O_3$) was measured by a Thermo Scientific Model 49i ozone analyzer, CO by a Thermo Scientific Model 48i carbon monoxide analyzer, $SO_2$ by an Ecotech EC 9850 sulfur dioxide analyzer, and $NO_2$ by a Thermo Scientific Model 42i $NO-NO_2-NOx$ analyzer. The meteorological data, including temperature, relative humidity (RH), precipitation, wind speed and wind direction, were measured by an automatic weather station (MAWS201, Vaisala, Vantaa, Finland) and a wind sensor (Vaisala Model QMW101-M2).

**2.3 Data analysis**

**2.3.1 Q-ACSM data analysis**

The mass concentrations and composition of $NR-PM_1$ were analyzed with the standard Q-ACSM data analysis software written in Igor Pro (WaveMetrics, Inc., OR, USA). Standard relative ionization efficiencies (RIEs) were used for organics, nitrate and chloride (i.e., 1.4 for organics, 1.1 for nitrate and 1.3 for chloride) (Ng et al., 2011a) and RIEs for ammonium (6.0) and sulfate (1.2) were derived from the IE calibrations. The particle collection efficiency (CE) was applied to correct for the particle loss at the vaporizer due to particle bounce, which is influenced by aerosol acidity, composition, and the aerosol water content. Given that aerosol was dried before entering into Q-ACSM and that ammonium nitrate mass fraction (ANMF) during the observation period was lower than 0.4, the composition dependent CE was estimated following the method described in Middlebrook et al. (2012).

**2.3.2 The Multilinear Engine (ME-2)**

PMF is a bilinear receptor model that represents an input data matrix as a linear combination of a set of factor profiles and their time-dependent concentrations (Paatero and Tapper, 1994). Factors typically correspond to unique sources and/or processes. This allows for a quantitative apportionment of bulk mass spectral time series into several factors through the minimization of a quantity $Q$, which





is the sum of the squares of the error-weighted residuals of the model. The PMF-AMS/ACSM analyses have been widely used for apportioning the sources of organic aerosol. However, in conventional PMF analyses, rotational ambiguity with limited rotational controls can lead to unclear factor resolution, especially in China where the emission sources are very complex and covariant during haze events. In contrast, the multi-linear engine (ME-2), used in this study, enables efficient exploration of the entire solution space and can direct the apportionment towards an environmentally-meaningful solution through the constraints of a subset of priori factor profiles or time series using the *a* value approach (Canonaco et al., 2013). The *a* value can vary between 0 and 1. An *a* value of 0.1 accounts for maximum ± 10% variability of each *m/z* signal of the final solution spectra that may differ from the anchor, implying that some *m/z* signals might increase while some might decrease.

The source finder (SoFi, Canonaco et al., 2013) tool version 4.9 for Igor Pro was used for ME-2 input preparation and result analysis. The number of factors resolved is determined by the user and the solutions of the model are not mathematically unique due to rotational ambiguity. It is, therefore, critical to study other parameters, e.g., the chemical fingerprint of the factor profiles, diurnal cycles, and time series of factors and external measurements, to support factor identification and interpretation (Canonaco et al., 2013; Crippa et al., 2014, Elser et al., 2016b).

**3 Results and discussion**

**3.1 Concentration and chemical composition of NR-PM$_1$**

Fig. 1 shows the time series of NR-PM$_1$ species, trace gases and meteorological conditions during the entire measurement period. The measured mass concentrations of NR-PM$_1$ for the entire campaign period ranged from a few µg m$^{-3}$ to 508.4 µg m$^{-3}$, with an average of 178 ± 101 µg m$^{-3}$. That was much higher than the wintertime/summertime concentrations measured in many other cities (see Table 1). The mass concentration of NR-PM$_1$ correlated strongly with that of PM$_{2.5}$ (R$^2$ = 0.76) with a regression slope of 0.72, indicating that NR-PM$_1$ represents a majority of PM$_{2.5}$ mass. The NR-PM$_1$ concentrations exceeded the Chinese PM$_{2.5}$ limit of 75 µg m$^{-3}$ for 90% of days during the measurement period, showing the severity of particulate air pollution at Shijiazhuang.





Similar to measurements at other urban sites, OA was the dominant fraction of NR-PM$_1$, with an
average of 50% (31-80%), followed by 21% of sulfate (4-36%), 12% of nitrate (2-26%), 11% of
ammonium (4-21%) and 6% of chloride (2-20%). The dominant contribution of organics in NR-PM$_1$
is also consistent with measurements from other urban sites in the BTH region during winter heating
seasons (see Table 1). Sulfate was the second largest contributor to NR-PM$_1$. The large fraction of
sulfate was likely associated with the large consumption of coal in Hebei province, i.e., 296 million
tons in 2014 were used in coal-fired power plants and steel industry (producing ∼11% of global steel
output in 2014). The enhancement of chloride fraction from >1-4% in other Chinese cities in summer
(see Table 1) to 6% in Shijiazhuang in winter (within the range of >2-7% in other Chinese cities in
winter, see Table 1) can be attributed to the substantial emissions from coal and/or biomass burning
activities.
Fig. 2a shows the diurnal variations of NR-PM$_1$ components, which were affected by the evolution of
the planetary boundary layer (PBL) height that governed the vertical dispersion of pollutants and by
the diurnal cycle of the emissions and atmospheric processes. The concentrations of pollutants
increased at night as a result of enhanced emissions from residential heating (in particular, for
organics and chloride) and a progressively shallower PBL. During daytime the PBL height was
developed by solar radiation and thus the pollutants became diluted resulting in the decrease of
organics, sulfate, ammonium and chloride in the afternoon. In contrast, the concentrations of nitrate
increased after sunrise but then kept rather constant throughout the afternoon, suggesting a strong
source or production of nitrate which offsets the dilution from PBL development. To minimize the
effects from PBL heights, data were normalized by ΔCO. CO is often used as an emission tracer to
account for dilution on timescales of hours to days because of its relatively long life time against the
oxidation by OH radicals (approximately one month) (Decarlo et al., 2010). After offsetting the PBL
dilution effect, sulfate, nitrate and ammonium showed clear increases from 7:00 to 15:00 (Fig. 2c),
indicating efficient daytime production of these secondary inorganic species. It should be noted that
the increase of nitrate (about 2 times, from ∼6 μg m$^{-3}$ ppm$^{-1}$ to ∼12 μg m$^{-3}$ ppm$^{-1}$) is slightly larger
than that of sulfate (about 1.6 times, from ∼11 μg m$^{-3}$ ppm$^{-1}$ to ∼17.5 μg m$^{-3}$ ppm$^{-1}$), indicating more
efficient photochemical production of nitrate than sulfate, given that the loss rate of sulfate could not



be higher than that of nitrate as nitric acid is semi-volatile and may be further lost by evaporation.
Also, the continuous increase of organics after sunrise suggested efficient photochemical production
of secondary organic aerosol (SOA).
**3.2 Sources of organic aerosol**
From the PMF analysis, we first examined a range of solutions with 3 to 8 factors. The solution that
best represents the data is the 5-factor solution (Fig. S1). The solutions with factor numbers more than
5 provide no new meaningful factors (see Fig. S2 and more details in the supplementary material).
Although the 5-factor solution can reasonably represent the data, HOA is still mixed with BBOA
because the HOA profile contains higher than expected contribution from m/z 60. In addition, COA
contains no signal at m/z 44, which might indicate a suboptimal splitting between the contributing
sources. To better separate HOA from BBOA, we constrained the HOA profile from Ng et al (2011b),
which is an average profile over 15 cities from China, Japan, Europe and the United States. Although
gasoline vehicles dominate in China while diesel vehicles dominate in Europe, HOA mass spectra do
not show significant variability when compared to different sites in China and Europe (Ng et al.,
2011b; Reyes et al., 2016; Bozzetti et al., 2017), indicating that traffic emissions from different types
of vehicles have similar profiles. To avoid the influences of other sources on COA, the COA profile
from Paris (Crippa et al., 2013) was used as a constraint because high similarities were found between
the COA profile from Paris and four COA profiles from different types of Chinese cooking activities
(He et al., 2010; Crippa et al., 2013). However, the constraint on HOA and COA profiles still seems
to sub-optimally resolve the apportionment of BBOA from CCOA, as one unconstrained factor
contains high contributions from both *m/z* 60 and PAH-related *m/z*'s (*m/z* 77, 91 and 115, as shown in
Fig. S3) which indicate the mixing between BBOA and CCOA. To separate BBOA and CCOA, we
constrained BBOA using the average of BBOA profiles from the 5-factor unconstrained PMF
solutions.
To explore the solution space, *a* value of 0-0.5 with an interval of 0.1 was used to constrain both the
HOA and COA reference profiles from literature while BBOA was constrained with *a* value of 0
because the BBOA profile was resolved from unconstrained PMF solution which is not expected to



1    vary significantly. 36 possible results were obtained by limiting a range of *a* values. Three criteria for

2    optimizing OA source appointment are as follows:

3        (1) *The diurnal pattern of COA*. The diurnal cycle of COA should have higher concentrations

4        during mealtime.

5        (2) *Minimization of m/z 60 in HOA*. The upper limit of m/z 60 in the HOA profile is 0.006, which

6        is the maximal fractional contribution derived from multiple ambient data sets in different

7        regions (mean + 2σ) (Ng et al., 2011b).

8        (3) *The rationality of unconstrained factors*. OOA should have abundant signal at *m/z* 44 and

9        contain much lower signals at PAH-related ion peaks compared to CCOA.

Nine solutions match the criteria above. The final time series and mass spectra are therefore the
averages of these 9 solutions. The diurnal variations of mass concentrations of the OA factors and
their PBL-corrected results are shown in Fig. 2b and d, respectively. The mass spectra and time series
of the OA factors and their correlation with external tracers are shown in Fig. 3. The relative
contributions of each OA source to the *m/z*'s are shown in Fig. S4. Potential source contribution
function (PSCF) analysis was also performed and the result is shown in Fig. S5.
OOA is characterized by high signals at *m/z* 44 ($CO_2^+$) and *m/z* 43 ($C_3H_7^+$ or $C_2H_3O^+$). OOA accounts
for 85% of *m/z* 44 signal, much higher than other OA sources. The time series of OOA is highly
correlated with that of sulfate ($R^2$=0.70), nitrate ($R^2$=0.75) and ammonium ($R^2$=0.76), confirming the
secondary nature of this factor. The diurnal cycle of OOA shows an increase from 7:00 to 11:00,
followed by a decrease in the afternoon due to the PBL evolution effect. After normalizing the PBL
effect, OOA increased continuously from 7:00 to 15:00, indicating the importance of photochemical
oxidation. This diurnal feature together with the PSCF results indicated that a large fraction of OOA
was produced locally and/or produced from the highly populated and industrialized surrounding areas,
consistent with the sulfate production discussed below.
The mass spectrum of CCOA is featured by prominent contributions of unsaturated hydrocarbons,
particularly PAH-related ion peaks (e.g., 77, 91, and 115). The CCOA profile shows a weaker signal





at $m/z$ 44 than that observed in Beijing (Hu et al., 2016a) and Lanzhou (Xu et al., 2016). This
difference can be caused by the difference in coal types, burning conditions and aging processes
(Zhou et al., 2016). CCOA accounts for 42-66% of PAH-related ion peaks, much higher than those in
other OA sources. This result suggested that the major source of PAHs was coal combustion in
wintertime Shijiazhuang. The average mass concentration of CCOA was 23.2 µg m$^{-3}$, which is higher
than that in Xi'an (10.1 µg m$^{-3}$) but is similar to that in Beijing (23.5 µg m$^{-3}$) observed in the same
winter (Elser et al., 2016a). CCOA showed distinct diurnal variations with low concentration down to
12.6 µg m$^{-3}$ during the day and high concentration up to 37.6 µg m$^{-3}$ at night, corresponding to 19%
and 35% of OA, respectively. The elevated CCOA concentrations at night suggested a large emission
from residential heating activities using coal as the fuel compounded by the shallow PBL. The
average contribution of CCOA to the total OA was 27%, which is consistent with studies in Beijing
and Handan (~160 km south to Shijiazhuang) where CCOA was found to be the dominant primary
OA (Elser et al., 2016a; Sun et al., 2016; Li et al., 2017). Given this large fraction of OA from coal
combustion, mitigating residential coal combustion is therefore of significant importance for
improving air quality in the BTH regions.
The BBOA mass spectrum is featured by prominent $m/z$ 60 (mainly $C_2H_4O_2^+$) and 73 (mainly
$C_3H_5O_2^+$) signals (He et al., 2010). These two ions ($C_2H_4O_2^+$ and $C_3H_5O_2^+$) are fragments of
anhydrous sugars produced from the incomplete combustion and pyrolysis of cellulose and
hemicelluloses (Alfarra et al., 2007; Lanz et al., 2007; Mohr et al., 2009). Consistently, BBOA
accounts for 50% of $m/z$ 60 and 56% of $m/z$ 73, much higher than those in other sources. In addition,
BBOA accounts for 9-27% of the PAH-related $m/z$'s (i.e., m/z 77, 91 and 115), lower than that in
CCOA but higher than those in other primary OA sources. This suggested that BBOA was also an
important PAH source in wintertime Shijiazhuang. A high correlation was found between the time
series of BBOA and that of chloride ($R^2$=0.75), the latter of which was suggested to be one of the
tracers of biomass burning. BBOA on average accounted for 17% of OA, which is higher than those
(9-12%) observed in Beijing during wintertime heating seasons (Elser et al., 2016a; Hu et al., 2016a;
Sun et al., 2016). The higher BBOA contribution in wintertime Shijiazhuang is likely associated with





widespread use of wood and crop residuals for heating and cooking in Shijiazhuang and surrounding
areas, as supported by the PSCF results (Fig. S5).
The COA profile is characterized by a high *m/z* 55/57 ratio of 2.7, much higher than that in non-
cooking POA (0.6-1.1) but within the range of 2.2-2.8 in COA profiles reported by Mohr et al.
(2012). COA shows a clear diurnal cycle with distinct peaks at lunch (between 11:00-13:00 local
time, LT) and dinner (between 19:00-21:00 LT) times. A small peak was also observed in the
morning between 06:00 and 07:00 LT, consistent with the breakfast time. COA on average accounted
for 16% of total OA with the highest contribution of 24% during dinner time.
The HOA mass spectrum is dominated by hydrocarbon ion series of $[C_nH_{2n+1}]^+$ and $[C_nH_{2n-1}]^+$
(Canagaratna et al., 2004; Mohr et al., 2009). The diurnal variation of HOA is featured by high
concentration at night, likely due to enhanced truck emissions (only allowed to drive on road from
23:00 to 6:00 LT) and shallow PBL at night. Similar diurnal cycles were found in wintertime Beijing
and Xi'an (Sun et al., 2016; Elser et al, 2016a). HOA, on average, accounted for 13% of total OA for
the entire observation period, which was higher than that in Beijing (9-10%) but lower than that in
Xi'an (15%) measured in the same winter (Elser et al., 2016a; Sun et al., 2016).
**3.3 Chemical nature and sources at different PM levels**
Fig. 4 shows the mass fractions of NR-PM$_1$ species and OA sources on reference days and extremely
polluted days. Here, the days with NR-PM$_1$ daily average mass concentration higher than the 75th
percentile (i.e., ≥238 µg m$^{-3}$) are denoted as the extremely polluted days and the rest of days as
reference days. The average concentration of NR-PM$_1$ was 310 µg m$^{-3}$ during extremely polluted
days, about 2 times higher than that during reference days (162 µg m$^{-3}$). The average concentration of
secondary inorganic aerosol was 65 µg m$^{-3}$ (40% of NR-PM$_1$ mass) during reference days and
increased to 143 µg m$^{-3}$ (46% of NR-PM$_1$ mass) during extremely polluted days. Secondary organic
aerosol also increased from 19 µg m$^{-3}$ (22% of OA) during reference days to 40 µg m$^{-3}$ (26% of OA)
during extremely polluted days. The enhanced mass concentrations (∼2 times) of both secondary
inorganic aerosol and secondary organic aerosol during extremely pollution days suggested strong
secondary aerosol production during pollution events. Such enhancement was likely compounded by



stagnant weather conditions (e.g., average wind speed was 0.9 m s$^{-1}$) and high RH of 69.4% which
facilitated the production and accumulation of secondary aerosol. Note that it was already very
polluted during the reference days with an average concentration of NR-PM$_1$ of 162 µg m$^{-3}$, which
may explain the relatively small increase in fractional contribution of secondary aerosol from
reference days to extremely polluted days.
Fig. 5a and b show the factors driving the pollution events by binning the fractional contribution of
each chemical species and OA source to total NR-PM$_1$ and OA mass, respectively. The data clearly
show that high pollution events are characterized by an increasing secondary fraction, reaching ~55%
at the highest NR-PM$_1$ mass bin (300-360 µg m$^{-3}$). In particular, from the lowest NR-PM$_1$ bin to the
highest NR-PM$_1$ bin, the fractional contribution increases from 14% to 25% for sulfate in NR-PM$_1$
and from 18% to 25% for OOA in OA, demonstrating the importance of secondary aerosol formation
in driving particulate air pollution (Huang et al., 2014; Elser et al., 2016; Wang et al., 2017). To
investigate the oxidation degree of sulfur at different NR-PM$_1$ mass, the sulfur oxidation ratio ($F_{SO4}$)
was calculated according to Eq. (1)

$$F_{SO_4^{2-}} = \frac{n\left[SO_4^{2-}\right]}{n\left[SO_4^{2-}\right] + n[SO_2]} \qquad (1)$$

where $n$ is the molar concentration. As can be seen from Fig. 6, $F_{SO4}$ shows a clear increase trend with
NR-PM$_1$ mass, increasing from 0.08 in the lowest mass bin to 0.21 in the highest mass bin. However,
the highest $F_{SO4}$ value is still much lower than that reported in previous studies, e.g., 0.62 in Xi'an
(Elser et al., 2016), suggesting low atmospheric oxidative capacity during the measurement period in
Shijiazhuang. This may also explain the relatively low OOA fraction (see Fig. 5b).
**3.4 Evolution of aerosol composition and sources at different RH levels**
Fig. 7a and b show the mass concentrations of the NR-PM$_1$ species and of the OA sources as a
function of RH, with RH bins of 10% increments. The absolute mass concentrations of secondary
inorganic species increased as RH increased from <60% to 90%, while chloride showed a decreasing
trend. Among the OA sources, OOA was significantly enhanced with RH increasing from <60% to



90%, while other OA sources did not show a clear trend. As RH increased gradually with the decrease
of wind speed (Fig. 6a), the development of stagnant weather conditions (including a shallower PBL)
promoted both the accumulation of pollutants and the formation of secondary aerosol (Tie et al.,
2016). To minimize the effects from PBL variations, the NR-PM$_1$ species and OA fractions were
normalized by the sum of the POA, as a surrogate of secondary aerosol precursors. The resulting
ratios were further normalized by the values at the first RH bin (<60%) for better visualization. As
shown in Fig. 7c, when RH increased from <60% to >90%, the normalized sulfate increased by a
factor of ∼2.5, suggesting the importance of aqueous-phase SO$_2$ oxidation in the formation of sulfate
at high RH. The enhancements for nitrate and ammonium were slightly lower (∼1.5) compared to that
sulfate, because NH$_4$NO$_3$ is thermal lability and its gas-particle partitioning is affected by both
temperature and RH. The importance of aqueous-phase chemistry is further supported by the increase
of $F_{SO4}$ as a function of RH (Fig. 6b). At RH <60%, $F_{SO4}$ was rather constant, with an average of 0.09,
indicating a low sulfur oxidation degree. At RH >60%, $F_{SO4}$ increased rapidly with the increase of
RH, reaching a maximal average of 0.18 at the last RH bin (90-100%). Note that the sulfur oxidation
degree at high RH (>60%) was much lower compared to those measured in Xi'an during the same
winter (average $F_{SO4}$ 0.62 at RH=90-100%, Elser et al., 2016a). The low sulfur oxidation degree
observed in Shijiazhuang (i.e., >80% of sulfur is still not oxidized) indicated insufficient atmospheric
processing and also suggested a large fraction of pollutants in Shijiazhuang was likely emitted locally
and/or transported from the heavily populated and industrialized surrounding areas. With a longer
atmospheric processing time in the downwind region, e.g., Beijing, higher secondary aerosol fractions
are expected, as observed in previous studies (e.g., Huang et al., 2014). Similar to sulfate, the
normalized OOA increased by a factor of ∼3 when RH increased from <60% to 90-100% (Fig. 7d).
The mass fraction of OOA increased from 29% to 41% when RH increased from 70% to 100%, while
POA contribution decreased correspondingly from 71% to 59% (Fig. 6d). These results support the
above discussion that aqueous-phase chemistry also plays an important role in the formation of OOA
under high RH conditions during haze pollution episodes.
**3.5 Primary emissions versus secondary formation**



Frequent changes between clean and polluted episodes were observed in this study. To get a better
insight into aerosol sources and atmospheric processes, 4 clean periods (C1-C4) with daily average
mass concentration of NR-PM$_1$ lower than the 25$^{th}$ percentile, 6 high-RH (>80%) polluted episodes
(H1-H6) and 4 low-RH (<60%) polluted episodes (L1-L4) with daily average mass concentration of
NR-PM$_1$ higher than the 75$^{th}$ percentile were selected for further analysis. As shown in the Fig. 8, the
chemical composition and sources differed during different episodes. The contributions of organics
showed a decreasing trend, from 54-64% during C1-C4 to 49-58% during L1-L4, and to 35-44%
during H1-H6, while the corresponding contributions of secondary inorganic species increased. This
indicated a notable production and accumulation of secondary inorganic aerosol during severe haze
pollution events. For example, the mass fraction of sulfate in NR-PM$_1$ was much higher during high
RH pollution events (H1-H6, 27-30%) compared to those during low RH pollution events (L1-L4, 11-
18%) and clean events (C1-C4, 11-17%). OOA also showed a much higher contribution to OA during
high RH pollution events (H1-H6, 29-50%) than during low RH pollution events (L1-L3, 17-26%)
and clean events (C1-C4, 10-34%). Interestingly, when comparing high RH and low RH pollution
events of similar PM levels (Fig. 8), secondary inorganic species and OOA dominated the particulate
pollution at high RH pollution events, similar to previous studies (e.g., Wang et al., 2017), while POA
dominated the particulate pollution at low RH and under stagnant conditions. These results highlight
the importance of meteorological conditions in driving particulate pollution.
Fig. 9 shows the evolution of aerosol species in two cases of different RH levels. The first case had
average RH <50% from 20-24 Jan (C2 and L3 episodes). The high wind speed (>6 m s$^{-1}$) from the
northwest before the L3 episode led to a significant reduction of air pollutants (the C3 episode, a
clean-up period). When the wind direction switched from northwest to 90°-270° sector and the wind
speed decreased to <3 m s$^{-1}$, the measured pollutants (except O$_3$ which was reacted out by increasing
NO emissions) started to build up. Specifically, NR-PM$_1$ showed a dramatic increase by a factor of 19
over the first 11 hours (from 20 Jan 16:00 to 21 Jan 3:00 LT) from 12 to 233 μg m$^{-3}$. In this process
POA contributed to an average 69% of NR-PM$_1$ mass. The other three processes were also
characterized by a rapid increase of NR-PM$_1$ mass (39-50 μg m$^{-3}$ h$^{-1}$) and high contribution of POA,
i.e., from 22 Jan 0:00-22 Jan 3:00, 22 Jan 16:00-22 Jan 20:00, and 23 Jan 12:00-23 Jan 19:00. Such



rapid increases in NR-PM$_1$ mass under low RH were associated with stagnant weather conditions
(e.g., low wind speed) which promoted the accumulation of pollutants. The second case had average
RH >80% from 5-8 Feb (H3 and H4 episodes). In this case, the wind speed was low (<3 m s$^{-1}$)
throughout the 4-day period. Under this very stagnant weather condition, POA accumulated
continuously (Fig. 9). However, different from the low-RH case, the concentration of secondary
species also showed continuous increases in this high-RH case. The enhancement of secondary
aerosol formation was likely driven by aqueous-phase chemistry at high RH level (Elser et al., 2016;
Wang et al., 2017) and the accumulation of pollutants under stagnant weather conditions (Tie et al.,
2017) which further promoted the formation of secondary species.
**4 Conclusions**
A Quadrupole Aerosol Chemical Speciation Monitor (Q-ACSM) was deployed in Shijiazhuang from
11 January to 18 February 2014 to investigate the chemical nature, sources and atmospheric processes
of fine particles in this heavily polluted city. The average mass concentration of NR-PM$_1$ was 178 ±
101 µg m$^{-3}$, much higher than the wintertime concentrations measured in many other cities. Organics
were the dominant composition (50%), followed by sulfate (21%), nitrate (12%), ammonium (11%)
and chloride (6%). As for the sources of OA, OOA (27%) and CCOA (27%) were on average the
most abundant sources, followed by BBOA (17%), COA (16%) and HOA (13%). The mass fractions
of secondary inorganic species and SOA increased with the increase of NR-PM$_1$ mass, suggesting the
importance of secondary formation in driving PM pollution. However, the low sulfur oxidation degree
and low OOA fraction indicated insufficient atmospheric oxidation capacity. Together with the
diurnal variations and PSCF results, these observations suggested that a large fraction of pollutants in
Shijiazhuang was most likely produced locally and/or transported from the heavily populated and
industrialized surrounding areas without sufficient atmospheric aging. Two different regimes were
found to be responsible for the high PM pollution in Shijiazhuang. At low RH under stagnant weather
conditions, the accumulation of primary emissions was the main culprit. In contrast, at high RH, the
enhanced formation of secondary aerosol through aqueous-phase chemistry was the main culprit. To
conclude, we found that in this highly polluted city in North China, (1) secondary formation is





important in high-PM episodes, (2) primary emissions are still important on an average basis, and (3)
meteorological conditions play an important part in pollutant accumulation and transformation. The
findings from this study thus suggest that (a) there are still opportunities for air pollution mitigation
by controlling direct emissions such as coal combustion, and (b) control on precursors (e.g., $NO_x$,
$SO_2$, and VOCs) for secondary formation, especially during high-PM episodes with unfavorable
meteorological conditions, can ease the situation substantially.

**5 Acknowledgement**

This research is supported by the National Science Foundation of China (No. 91644219 and
41675120), and EPA-Ireland (AEROSOURCE, 2016-CCRP-MS-31).

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



1    Table 1. The fine PM mass concentrations and fractional contribution of different composition in

2    different locations.

| City | Season | NR-PM$_1$ (μg m$^{-3}$) | OA % | SO$_4^{2-}$ % | NO$_3^-$ % | NH$_4^+$ % | Cl$^-$ % | Ref. |
|------|--------|------|------|------|------|------|------|------|
| Beijing | Winter, 2010 | 60 | 54 | 14 | 11 | 12 | 9 | Hu et al., 2016a |
| Beijing | Winter, 2011 | 59 | 51 | 13 | 17 | 14 | 5 | Sun et al., 2015 |
| Beijing | Winter, 2012 | 66.8 | 52 | 14 | 16 | 13 | 5 | Sun et al.,2013 |
| Beijing | Winter, 2012 | 79 | 52 | 17 | 14 | 10 | 7 | Wang et al., 2015 |
| Beijing | Winter, 2013 | 77 | 50 | 19 | 16 | 12 | 3 | Sun et al., 2014 |
| Beijing | Winter, 2013 | 13.0 | 52 | 17 | 14 | 10 | 7 | Jiang et al., 2015 |
| Beijing | Winter, 2013 | 64 | 60 | 15 | 11 | 8 | 6 | Sun et al., 2016 |
| Beijing | Winter, 2014 | 75[a] | 56 | 16 | 10 | 7 | 11 | Elser et al., 2016 |
| Beijing | Summer, 2011 | 80 | 32 | 28 | 21 | 17 | 2 | Hu et al., 2016a |
| Beijing | Summer, 2012 | 52 | 41 | 14 | 25 | 17 | 3 | Sun et al., 2015 |
| Lanzhou | Winter, 2014 | 57.3 | 55 | 13 | 18 | 11 | 3 | Xu et al., 2016 |
| Lanzhou | Summer, | 24 | 53 | 18 | 11 | 13 | 5 | Xu et al., 2014 |





| | | | | | | | |
|---|---|---|---|---|---|---|---|
| | 2012 | | | | | | |
| Ziyang | Winter, 2012 | 60 | 40 | 24 | 15 | 17 | 4 | Hu et al., 2016b |
| Handan | Winter, 2015 | 178 | 47 | 16 | 15 | 13 | 9 | Li et al., 2017 |
| Shenzhen | Autumn, 2009 | 38.3 | 46 | 29 | 12 | 11 | 2 | He et al., 2011 |
| Shanghai | Summer, 2010 | 27 | 31 | 36 | 17 | 14 | 2 | Huang et al., 2012 |
| Nanjing | Summer, 2013 | 36.8 | 42 | 14 | 24 | 19 | 1 | Zhang et al., 2015b |
| Hong Kong | Winter, 2012 | 14.5 | 33 | 40 | 10 | 16 | 1 | Li et al., 2015b |
| Hong Kong | Summer, 2011 | 8.7 | 26 | 56 | 3 | 15 | 0.1 | Li et al., 2015b |
| Paris | Winter, 2010 | 16.7 | 35 | 16 | 33 | 15 | 1 | Crippa et al., 2013 |
| Fresno, Califonia | Winter, 2010 | 11.8 | 67 | 3 | 20 | 8 | 2 | Ge et al., 2012 |
| Shijiazhuang | Winter, 2014 | 178 | 50 | 21 | 12 | 11 | 6 | This study |

1  [a]NR-PM$_{2.5}$




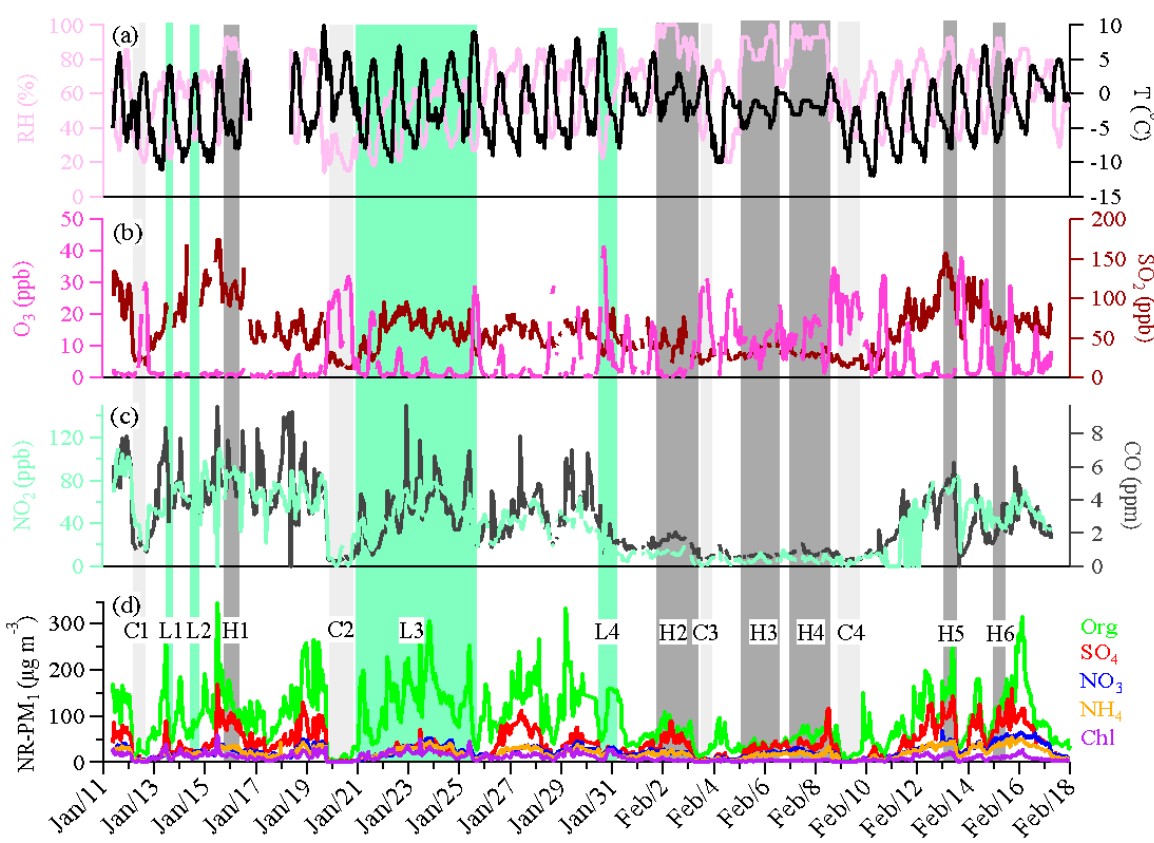

**Fig. 1 Time series of relative humidity and temperature(a), O₃ and SO₂ (b), NO₂ and CO (c),**

**and the NR-PM₁ species (d) during the observation period. 6 high-RH (>80%) polluted episodes**

**(H1-H6), 4 low-RH (<60%) polluted episodes (L1-L4), and 4 clean episodes (C1-C4) are marked**

**for further discussion.**





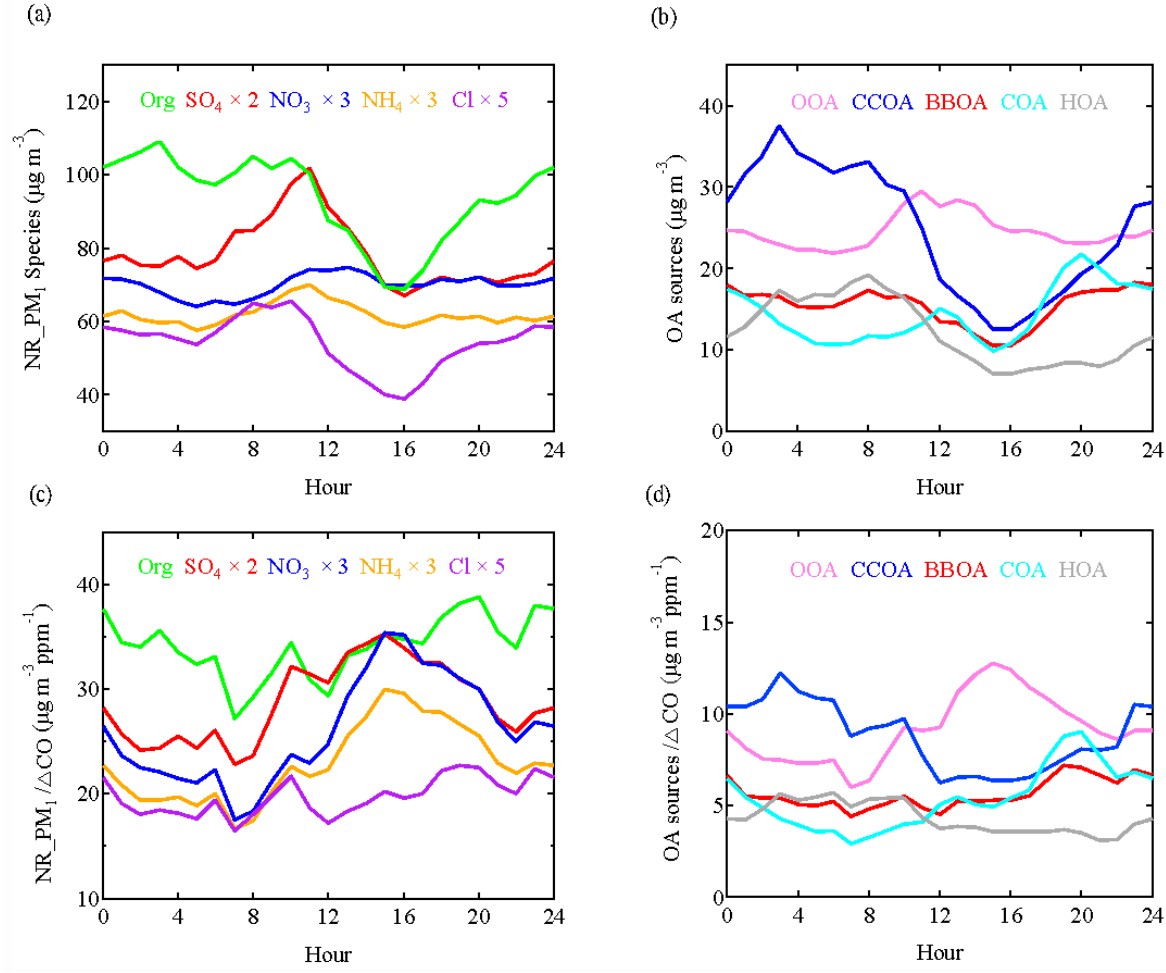

Fig. 2. Diurnal variations of NR-PM$_1$ composition (a), OA sources (b), NR-PM$_1$ species/ΔCO (C) and OA sources/ΔCO (d).





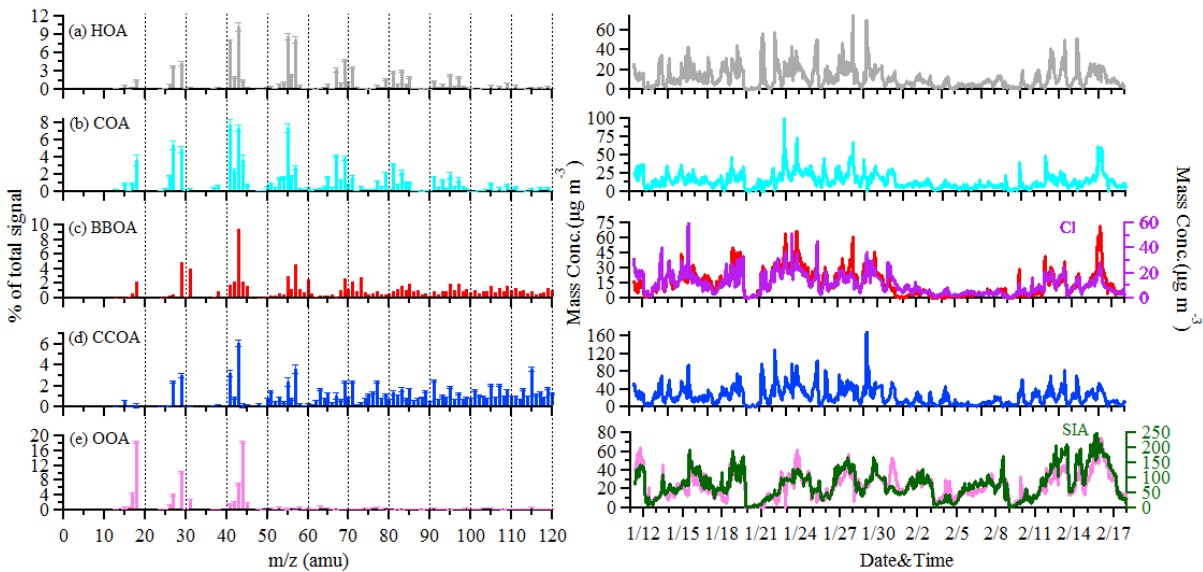

**Fig. 3 Mass spectrums (left) and time series (right) of five OA sources. Error bars of mass spectrums represent the standard deviation of each *m/z* over all accepted solutions.**



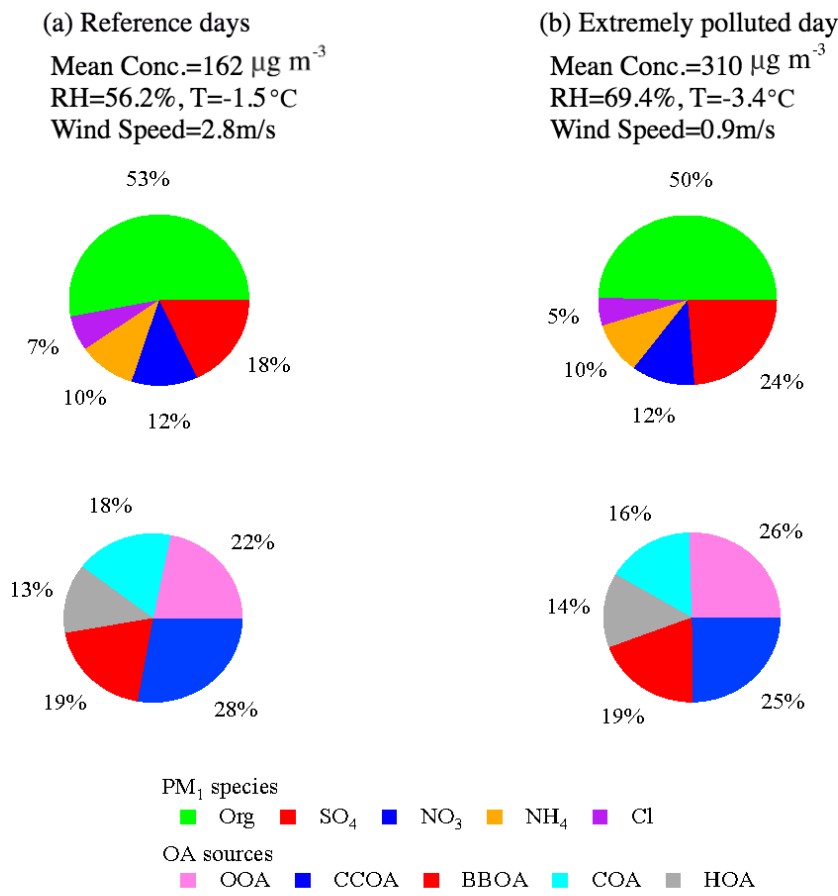

**Fig. 4. Relative contributions of NR-PM$_1$ species and OA sources (OOA, CCOA, BBOA, COA and HOA) in reference days (a) and extremely polluted days (b). Extremely polluted days are defined as the daily average mass concentration of NR-PM$_1$ higher than the 75$^{th}$ percentile (237.3 μg m$^{-3}$) and the rest refers to the reference days. Data in the Spring Festival is excluded to eliminate the influence from a change of emission patterns in the holiday.**



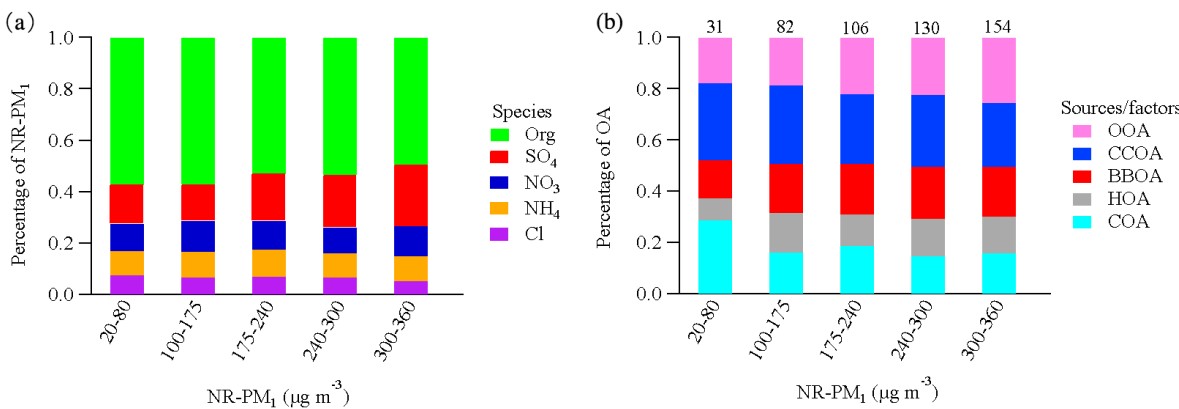

**Fig. 5. Relative contributions of NR-PM$_1$ species (a) and OA sources (b) as a function of daily average NR-PM$_1$ mass concentrations. The numbers above the bars refer to the OA mass concentration ($\mu g\ m^{-3}$). Data in the Spring Festival is excluded to eliminate the influence from the change of emission patterns in the holiday.**





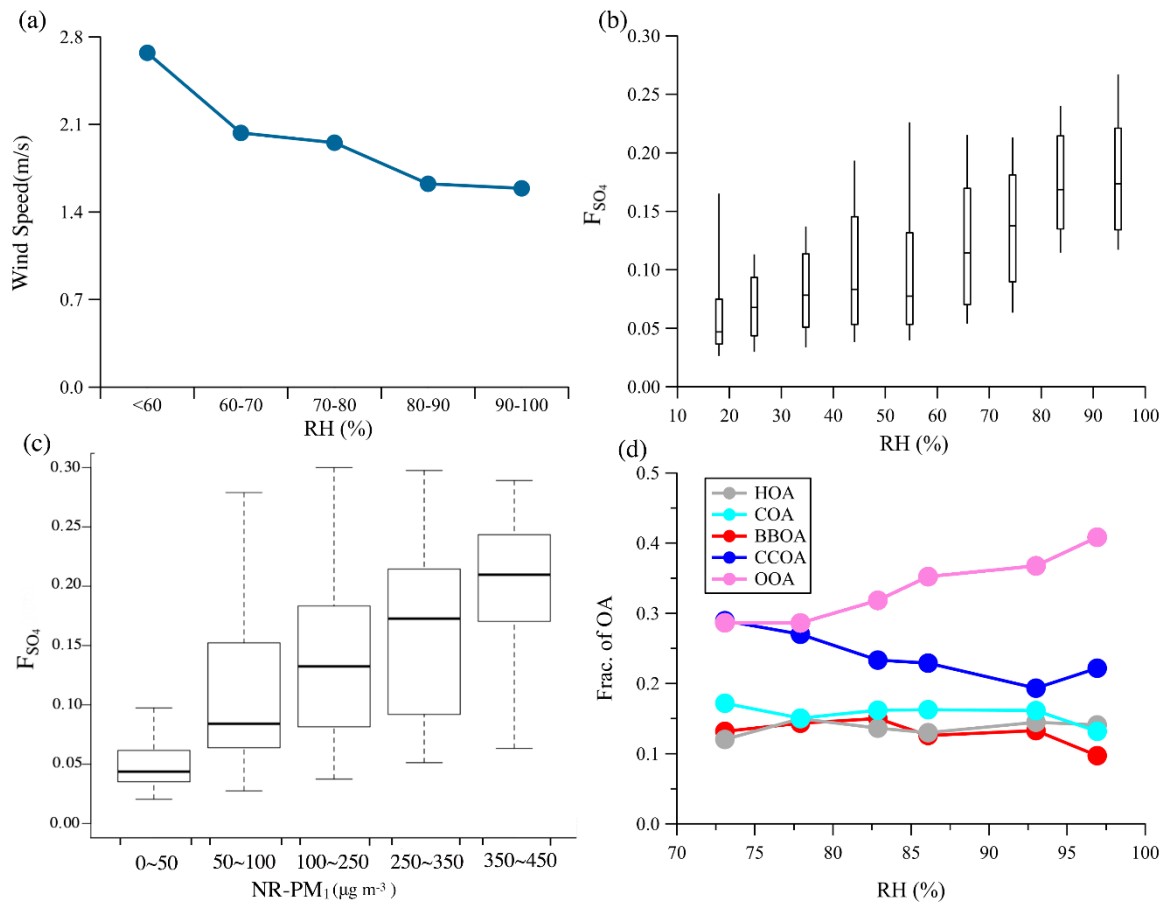

2  **Fig. 6. Variations of wind speed as a function of RH (a), $F_{SO4}$ as a function of RH (b) and of the**

3  **NR-PM$_1$ mass concentrations (c), and the mass fraction of organic as a function of RH (d).**



**Fig. 7. The average mass concentration of NR-PM$_1$ species (a) and OA sources (b) as a function of RH. The average mass concentration of NR-PM$_1$ species (c) and OA sources (d) normalized to the sum of primary sources (HOA, COA, BBOA, and CCOA) as a function of RH. All ratios are further normalized to the values at the first RH bin (<60%) for the better illustration.**





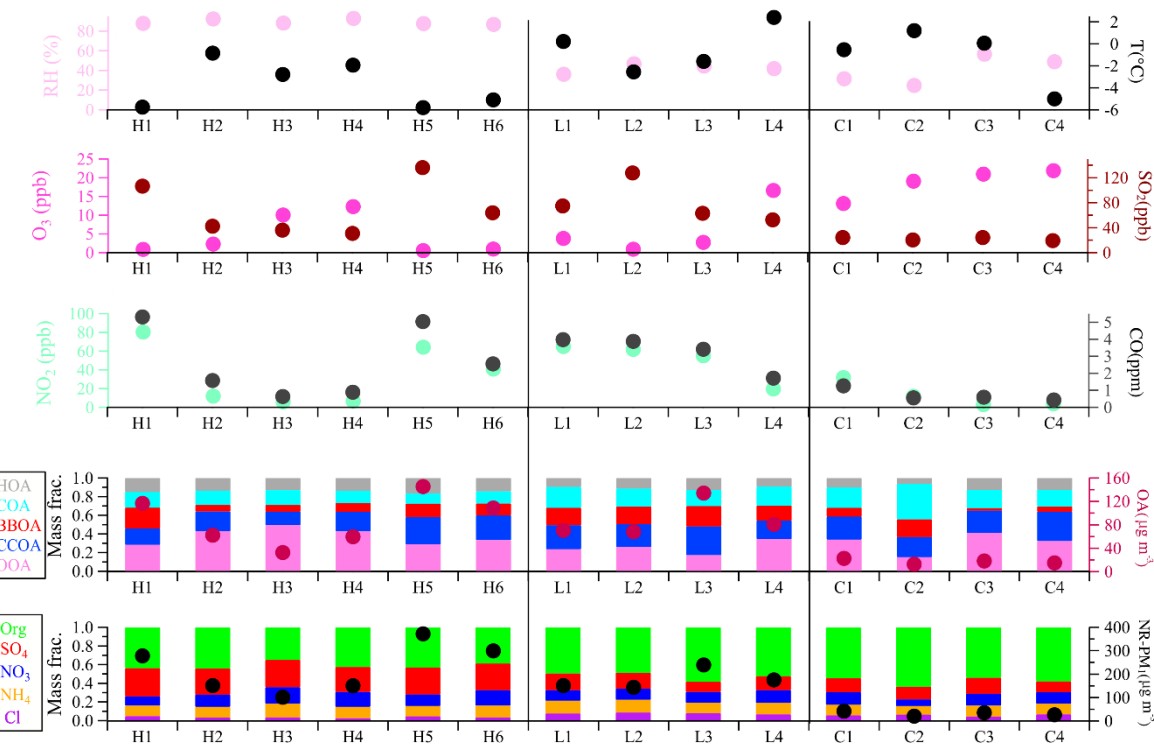

**Fig. 8. Summary of relative humidity and temperature, gaseous species, organic sources and**

**NR-PM₁ chemical composition for high-RH (H1-H6) polluted, low-RH (L1-L4), and clean (C1-**

**C4) episodes.**




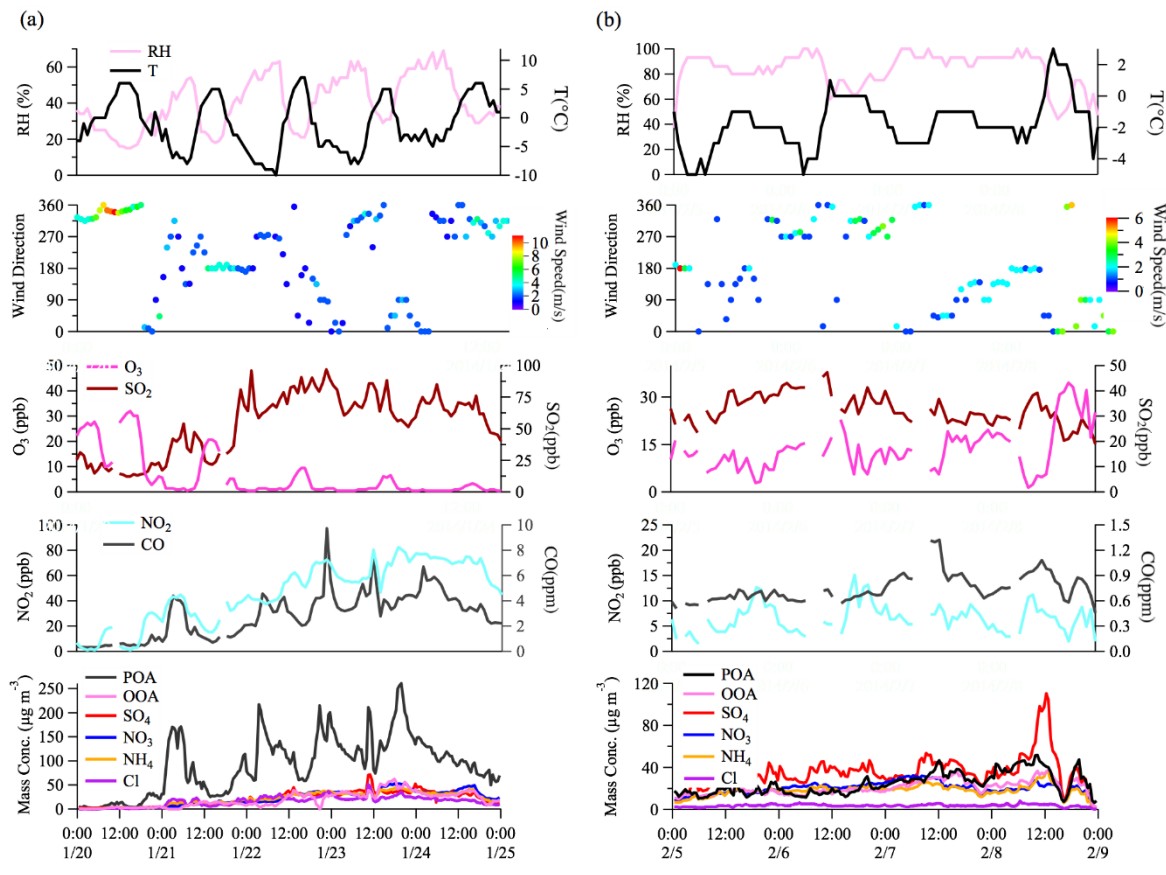

**Fig. 9. Time series of meteorological factors (relative humidity, temperature, wind speed and wind direction), gaseous species, OA factors and NR-PM$_1$ chemical composition for the first period (average RH <50%) (a) and the second period (average RH>80%) (b).**