# Peer review of "Primary emissions versus secondary formation of fine particulate matter in the top polluted"

_Atmospheric Chemistry and Physics, 2018_

## Referee Comment (RC1) · Anonymous Referee #1 · 10 Sep 2018

Review of "Primary emissions versus secondary formation of fine particulate matter in the top polluted city, Shijiazhuang, in North China" by Huang et al.

This manuscript reports the measurements of non-refractory fine particulate matter (NR-PM$_1$) in Shijiazhuang, China, using a quadrupole aerosol chemical speciation monitor (Q-ACSM). Positive matrix factorization (PMF) analysis is performed for source apportionment of organic aerosol (OA). It is found that on average primary emissions are the major source of NR-PM$_1$, but secondary pollutants via aqueous phase reactions play a more important role in polluted events. The data analysis is routine and the conclusions are broadly consistent with many previous studies in the same region. Overall, I recommend publication after major revisions.

Major Comments

1.      Page 11 Line 4-5. The coal combustion OA (CCOA) is identified based on PAH-related ion peaks in its mass spectrum and the assumption that PAH is mainly from coal combustion. So it is problematic to draw the conclusion that "the major source of PAHs was coal combustion".

2.      Page 11 Line 5-6. As the CCOA concentration is similar between Beijing and Shijiazhuang, does it suggest that the major source of CCOA in Beijing is from local emissions? In the introduction (Page 4 Line 5-15), it is mentioned that previous studies estimate a large fraction of PM in Beijing arising from regional transport. If so, one would imagine that the CCOA concentration is higher in surrounding area, like Hebei, than Beijing.

3.      Section 3.4. The increase in SO$_4$/POA and OOA/POA is largely due to the decrease in POA at high RH (90-100% bin). What causes the decrease in POA concentration? Precipitation? Have the precipitation events been excluded from the analysis?

        In Figure 7, the light blue should be COA, instead of CCOA.

4.      Could the authors provide more explanations regarding why POA is important in low RH polluted days, but SOA is important in high RH polluted days? I would imagine that the POA emissions do not vary with RH. Then where does the POA go in high RH polluted days?

        According to Figures 1 and 8, there is larger variation in the OA concentration between the six high RH events than the variation between four low RH events. H3, H5, and L3 seem to be outliers.

Minor Comments

1.	Page 2 Line 21. "Concentration" should be plural.

2.	Page 4 Line 21. What's the diurnal trend of delta_CO? Are there rush hour peaks?

3.	Page 9 Line 5-24. From my understanding, these paragraphs discuss results from unconstrained PMF, right? If so, please be more specific.

4.	Page 12 Line 27. Change "compounded" to "confounded".

---

## Referee Comment (RC2) · Anonymous Referee #2 · 14 Nov 2018

The authors present a valuable dataset from aerosol mass spectrometry measurements in Shijiazhuang, Hebei, China during the winter of 2014. They have applied the ME-2 approach for more accurate identification of sources. In my assessment the article will be suitable for publication in ACP after a few minor points are addressed:

- This article is closely related to Zhu et al. AMT 2018, which also presents ME-2 analysis of organic aerosol mass spectrometry data from China. The two articles share some of the same authors, and Zhu et al. was available online in final form prior to this article appearing in APCD. That article should be cited and discussed as appropriate in this manuscript.

- Low FSO4 does not necessarily indicate low oxidative power in the atmosphere. SO2 oxidation may occur through a number of multiphase pathways, and other parameters such as aerosol or cloudwater pH are more likely to influence FSO4. Please amend the statements regarding the implications of low FSO4 throughout the manuscript in light of this caveat.

- The 1.36e6 premature deaths figure is out of date at this point - please refer to a more recent study such as Cohen et al. Lancet 2017 or Burnett et al. PNAS 2018.

- Can anything be inferred from these results regarding the impacts of regional pollution on PM in Beijing?

- page 5 line 8: 15 March of the next year

- page 5 line 9: 2013-2014

- page 14 line 10: thermally labile

- the Conclusion section is too repetitive, please revise to be more distinct from the Abstract and other sections of the paper

---

## Author Comment (AC1) · 16 Jan 2019

The authors thank the editor and referees to review our manuscript and particularly for the valuable comments and suggestions that have significantly improved the manuscript. We provide below point-by-point responses to the referees' comments. We also have made most of the changes suggested by the referees in the revised manuscript.

Referee #1

This manuscript reports the measurements of non-refractory fine particulate matter (NR-PM1) in Shijiazhuang, China, using a quadrupole aerosol chemical speciation monitor (Q-ACSM). Positive matrix factorization (PMF) analysis is performed for source apportionment of organic aerosol (OA). It is found that on average primary emissions are the major source of NR-PM1, but secondary pollutants via aqueous phase reactions play a more important role in polluted events. The data analysis is routine and the conclusions are broadly consistent with many previous studies in the same region. Overall, I recommend publication after major revisions.

Major Comments

1.  Page 11 Line 4-5. The coal combustion OA (CCOA) is identified based on PAH-related ion peaks in its mass spectrum and the assumption that PAH is mainly from coal combustion. So it is problematic to draw the conclusion that "the major source of PAHs was coal combustion".

Response: PAHs are emitted from coal combustion, biomass burning and vehicular emissions, and PAH-related ions are assigned in these sources in the PMF source apportionment. We did not make the assumption that PAH is mainly from coal combustion, and the assignment is from PMF model runs. For example, in our previous study (Elser et al., 2016a), from PMF study we quantified that the main source of PAHs was biomass burning in Xi'an and was coal combustion in Beijing during severe haze events. In the present study, the PMF model results show that CCOA accounts for 42-66% of PAH-related ions and BBOA accounts for 9-27% of PAH-related ions. We therefore concluded that "this result suggested that the major source of PAHs was coal combustion in wintertime Shijiazhuang."

2.  Page 11 Line 5-6. As the CCOA concentration is similar between Beijing and Shijiazhuang, does it suggest that the major source of CCOA in Beijing is from local emissions? In the introduction (Page 4 Line 5-15), it is mentioned that previous studies estimate a large fraction of PM in Beijing arising from regional transport. If so, one would imagine that the CCOA concentration is higher in surrounding area, like Hebei, than Beijing.

Response: The average mass concentration of CCOA was similar between Shijiazhuang and Beijing, but they were campaign-averaged concentrations including clean and haze events. In Beijing, the clean periods were generally associated with northerly and northwesterly winds (i.e., from the clean mountain area), while haze extremes were related to southerly winds (i.e., from the polluted southern Hebei including Shijiazhuang). We revisited our data and found that, during haze extremes in the same winter, the average CCOA concentration was 77.5 μg m$^{-3}$ in Shijiazhuang, much higher than that in Beijing (48.2 μg m$^{-3}$).

In the revised manuscript, we have added "Nevertheless, during haze extremes, the average CCOA concentration was 77.5 μg m-3 in Shijiazhuang, much higher than that in Beijing (48.2 μg m-3, Elser et al., 2016a)."

3.  Section 3.4. The increase in SO4/POA and OOA/POA is largely due to the decrease in POA at high RH (90-100% bin). What causes the decrease in POA concentration? Precipitation? Have the precipitation events been excluded from the analysis? In Figure 7, the light blue should be COA, instead of CCOA.

Response: We thanks the referee to point out the flaws. Indeed, the decrease in POA at the RH 90-100% bin was caused by snow. We have excluded the data from snow events in the revised manuscript, and Figure 7 has been updated accordingly (see below). Also, we have changed the light blue from CCOA to COA.

[Figure]

4.  Could the authors provide more explanations regarding why POA is important in low RH polluted days, but SOA is important in high RH polluted days? I would imagine that the POA emissions do not vary with RH. Then where does the POA go in high RH polluted days?

Response: This conclusion was drawn based on the mass fraction contribution. During high RH pollution events, more SOA and SIA were formed and therefore their fractional contribution increased. The POA concentrations are determined by both emissions and meteorological conditions.

In the revised manuscript, we have added "…at high RH pollution events likely due to

enhanced secondary formation,"; and "The concentrations of POA are determined by both emissions and meteorological conditions. The different significance of primary aerosol and secondary aerosol in low and high RH pollution events highlights the importance of meteorological conditions in driving particulate pollution."

According to Figures 1 and 8, there is larger variation in the OA concentration between the six high RH events than the variation between four low RH events. H3, H5, and L3 seem to be outliers.

Response: The OA concentrations are determined by emissions, secondary formation, and meteorological conditions. The larger variation in the OA concentrations/sources between the six high RH events is most likely due to SOA formation. The relatively smaller variation in the OA concentrations/sources between the four low RH events is likely related to stagnant air which facilitates the accumulation of particles.

For H3, H5, and L3 events, each spanned relatively long period (from ~12 hr to ~108 hr). They are real measurement data, not outliers.

Minor Comments

1. Page 2 Line 21. "Concentration" should be plural.
Response: Change made.

2. Page 8 Line 21. What's the diurnal trend of delta_CO? Are there rush hour peaks?
Response: The diurnal trend of delta_CO is shown below. There are rush hour peaks.

[Figure]

3. Page 9 Line 5-24. From my understanding, these paragraphs discuss results from unconstrained PMF, right? If so, please be more specific.
Response: From Line 5-11, we discussed the standard PMF results (without constraints); from Line 11-24, we explained which factors were constrained.

4. Page 12 Line 27. Change "compounded" to "confounded".
Response: Change made.

Referee #2
The authors present a valuable dataset from aerosol mass spectrometry measurements in Shijiazhuang, Hebei, China during the winter of 2014. They have applied the ME-2

approach for more accurate identification of sources. In my assessment the article will be suitable for publication in ACP after a few minor points are addressed:

- This article is closely related to Zhu et al. AMT 2018, which also presents ME-2 analysis of organic aerosol mass spectrometry data from China. The two articles share some of the same authors, and Zhu et al. was available online in final form prior to this article appearing in APCD. That article should be cited and discussed as appropriate in this manuscript.
Response: In the revised manuscript, the article from Zhu et al., 2018 has been cited properly.

- Low FSO4 does not necessarily indicate low oxidative power in the atmosphere. SO2 oxidation may occur through a number of multiphase pathways, and other parameters such as aerosol or cloud water pH are more likely to influence FSO4. Please amend the statements regarding the implications of low FSO4 throughout the manuscript in light of this caveat.
Response: We agree with the reviewer that sulfate formation may occur through different pathways, including gas-phase oxidation of $SO_2$ by OH and multiphase reactions of $SO_2$ with dissolved ozone, hydrogen peroxide, organic peroxides, $NO_2$, and OH via catalytic or non-catalytic pathways involving mineral oxides. However, these reactions are oxidation reactions, and here we used FSO4 to represent the oxidation ratio of sulfate. The low oxidation ratio of FSO4 suggests low atmospheric oxidative capacity. We think such statement is appropriate. Certainly, the absolute mass concentration of sulfate may also be affected by other parameters including aerosol liquid water content, aerosol or cloud water pH.
In the revised manuscript, we have added "Certainly, it should be noted that the mass concentration of sulfate may also be affected by other parameters including aerosol liquid water content, aerosol or cloud water pH, besides atmospheric oxidative capacity."

- The 1.36e6 premature deaths figure is out of date at this point - please refer to a more recent study such as Cohen et al. Lancet 2017 or Burnett et al. PNAS 2018.
Response: We have made change "1.1 million deaths in 2015 in China", following the results in Cohen et al., 2017.

- Can anything be inferred from these results regarding the impacts of regional pollution on PM in Beijing?
Response: This study was mainly concentrated on the chemical characteristics, primary emissions and secondary formation processes of aerosol in Shijiazhuang, which is located in the west to Beijing. It certainly will be of interest to investigate the impacts of regional pollution on PM in Beijing, and such studies would need synchronous measurements in Beijing and Shijiazhuang.

- page 5 line 8: 15 March of the next year
Response: Change made.

- page 5 line 9: 2013-2014
Response: Change made.

- page 14 line 10: thermally labile
Response: Change made.

- the Conclusion section is too repetitive, please revise to be more distinct from the Abstract and other sections of the paper
Response: We have shortened the conclusion section.